# On the analysis of genetic association with long-read sequencing data

Gengming He[1,2], Stephen W. Scherer[2,3,4], Lisa J. Strug[1,2,3,5]*

**1** Biostatistics Division, Dalla Lana School of Public Health, University of Toronto, Toronto, Ontario, Canada, **2** Program in Genetics and Genome Biology, The Hospital for Sick Children, Toronto, Ontario, Canada, **3** The Centre for Applied Genomics, The Hospital for Sick Children, Toronto, Ontario, Canada, **4** McLaughlin Centre and Department of Molecular Genetics, University of Toronto, Toronto, Ontario, Canada, **5** Departments of Statistical Sciences and Computer Science, University of Toronto, Toronto, Ontario, Canada

* lisa.strug@utoronto.ca

**Data availability statement:** The 10XG data referenced in this study are deposited in the

## Abstract

Long-read sequencing (LRS) technologies have enhanced the ability to resolve complex genomic architecture and determine the 'phase' relationships of genetic variants over long distances. Although genome-wide association studies (GWAS) identify individual variants associated with complex traits, they do not typically account for whether multiple associated signals at a locus may act in *cis* or *trans*, or whether they reflect allelic heterogeneity. As a result, effects that arise specifically from phase relationships may remain hidden in analyses using short-read and microarray data. While the advent of LRS has enabled accurate measurement of phase in population cohorts, statistical methods that leverage phase in genetic association analysis remain underdeveloped. Here, we introduce the Regression on Phase (RoP) method, which directly models *cis* and *trans* phase effects between variants under a regression framework. In simulations, RoP outperforms genotype interaction tests that detect phase effects indirectly, and distinguishes in-*cis* from in-*trans* phase effects. We implemented RoP at two cystic fibrosis (CF) modifier loci discovered by GWAS. At the chromosome 7q35 *trypsinogen* locus, RoP confirmed that two variants contributed independently (allelic heterogeneity). At the *SLC6A14* locus on chromosome X, phase analysis uncovered a coordinated regulatory mechanism in which a promoter variant modulates lung phenotypes in individuals with CF when acting in *cis* with a lung-specific enhancer (E2765449/enhD). This coordinated regulation was confirmed in functional studies. These findings highlight the potential of leveraging phase information from LRS in genetic association studies. Analyzing phase effects with RoP can provide deeper insights into the complex genetic architectures underlying disease phenotypes, ultimately guiding more informed functional investigations and potentially revealing new therapeutic targets.

Canadian Cystic Fibrosis registry (https://cysticfibrosis.ca/canadian-cystic-fibrosis-registry). Access to the data for research and clinical study purposes is available upon formal request to Cystic Fibrosis Canada (cfregistry@cysticfibrosis.ca), subject to review and approval from the Registry Review Panel. The R code for simulations is available at https://github.com/tony891210/RoP-Simulation. The R package *RegPhase* is available on GitHub at https://github.com/strug-hub/RegPhase.

**Funding:** LJS is funded by the peer-reviewed Cystic Fibrosis Canada 2022 Basic and Clinical Research Grant (https://cysticfibrosis.ca/, 1009794), jointly funded by CF Canada and Canadian Institutes of Health Research Institute of Circulatory and Respiratory Health (CIHR-ICRH) (https://cihr-irsc.gc.ca/e/8663.html, FRN: BCG 187014); Cystic Fibrosis Canada Grant (https://cysticfibrosis.ca/, 608828); Cystic Fibrosis Foundation (https://www.cff.org/, STRUG17PO); the Canadian Institutes of Health Research (https://cihr-irsc.gc.ca/e/193.html, FRN-167282); and by the Government of Canada through Genome Canada (https://genomecanada.ca/), Ontario Genomics Institute (https://www.ontariogenomics.ca/, OGI-148), and The Centre for Applied Genomics (https://www.tcag.ca/, Genome Facility Funding Opportunity, Technology Development). This research was undertaken, in part, thanks to funding from the Canada Research Chairs Program to LJS (https://www.chairs-chaires.gc.ca/home-accueil-eng.aspx), who holds the Canada Research Chair in Genome Data Science. GH is funded by CANSSI Ontario (https://canssiontario.utoronto.ca/) as a fellow of STAGE (Strategic Training for Advanced Genetic Epidemiology) and by the Data Sciences Institute at the University of Toronto (https://datasciences.utoronto.ca/). The funders had no role in study design, data collection and analysis, decision to publish, or preparation of the manuscript.

**Competing interests:** The authors have declared that no competing interests exist.

## Author summary

Traditional genetic association studies typically link individual genetic variants to diseases but often neglect how variants may jointly affect outcomes based on their arrangement across maternal and paternal chromosomes, known as phase relationships. Understanding phase effects is essential for uncovering the mechanisms underlying complex diseases. Recent advances in long-read sequencing technology allow precise measurement of phase relationships over extensive chromosome regions; however, statistical methods for analyzing these effects remain limited. We developed a novel statistical approach called Regression on Phase (RoP) to directly assess these complex genetic interactions. Our simulation studies demonstrated that RoP effectively identifies effects dependent on specific phase arrangements. Applying RoP to genetic variants contributing to cystic fibrosis (CF) revealed phase-dependent mechanisms affecting CF-related lung disease, which were missed by traditional methods. Analyzing phase effects with RoP can advance our understanding of disease mechanisms, guide future functional studies, and ultimately support the development of personalized medicine.

## Introduction

Long-read (LRS) sequencing technologies, such as those by Pacific Biosciences (PacBio) and Oxford Nanopore, have significantly advanced genomic research by producing single reads spanning tens to hundreds of kilobase pairs. These platforms address limitations inherent in traditional short-read sequencing by enabling accurate assembly of repetitive and complex genomic regions and providing precise phase information for DNA variants across extensive genomic intervals. Previously, high costs limited the application of LRS to small cohorts, constraining its utility in genetic association studies of complex traits. However, recent developments in LRS, such as PacBio's High-Fidelity (HiFi) sequencing technology [1], have led to improvements in read accuracy, throughput, and cost-effectiveness. The 10X Genomics (10XG) linked-read technology, on the other hand, measures long-range phase information (phase block N50 = 4.39 Mb [2]) with standard short-read platforms by bar-coding reads from the same chromosome and post hoc reassembly [3]. These advancements enable the sequencing of larger disease cohorts and expand the potential of LRS to link novel genetic variation and complex alleles to disease outcomes by association analysis. Despite these technological breakthroughs, a critical methodological gap remains, that is, how to fully leverage phase information in genetic association analyses to better understand the mechanisms underlying complex genetic diseases.

Phase refers to the arrangement of alleles across the two homologous chromosomes inherited from each parent, determining whether genetic variants are co-inherited in-*cis* (on the same chromosome) or in-*trans* (on opposite chromosomes) (Fig 1A). Phase effects can profoundly influence how combinations of genetic variants impact phenotypes across various diseases [4]. For example, consider cystic fibrosis (CF), which is a recessive genetic disorder caused by loss-of-function variants in *CFTR*. The 5T variant in the intron 8 poly-T tract of *CFTR* leads to splicing alterations and causes disease when in-*cis* with the missense variant D1152H, and in-*trans* with a second CF-causing variant [5,6]. Genes beyond *CFTR*, known as CF modifier genes, modulate disease severity and progression. While genome-wide association studies (GWAS) leveraging millions of single-SNP association tests have successfully identified genomic regions associated with complex traits, such as genetic modifiers of lung and intestinal disease severity in CF, they provide limited insight into the genetic mechanisms

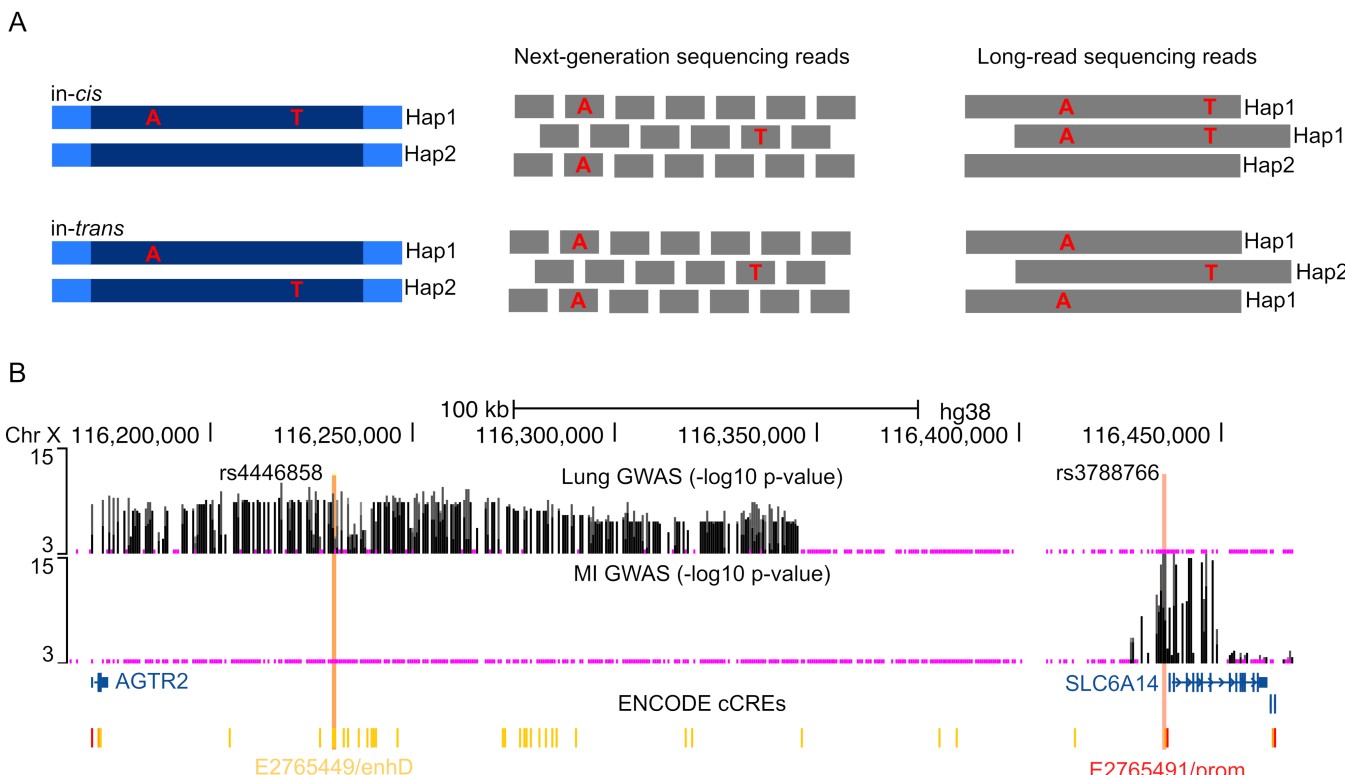

**Fig 1. Long-read sequencing clarifies phase relationships.** (A) A schematic of how *cis* (top) and *trans* (bottom) relationships between two genetic variants can be difficult to infer with conventional short-read sequencing. Short-read data (read length typically ≤ 300 bp) rarely span both variants within one continuous read, and their phasing often remains ambiguous. Long-read sequencing can capture both variants on the same read, enabling direct determination of *cis* or *trans* relationships. (B) The meconium ileus (MI) and lung traits GWAS associations (shown as $-log_{10}$ p-values) across a ~200 kb region near the *SLC6A14* locus on chromosome X (hg38). The putative functional variants from the MI (rs3788766) and lung (rs4446858) GWAS that overlap with ENCODE candidate cis-regulatory elements (cCREs) are indicated by vertical orange lines.

driving these associations. LRS technologies provide the opportunity to incorporate phase information and complex variation, enabling a more comprehensive understanding of complex traits by elucidating whether variants interact functionally based on their chromosomal configuration, rather than treating single-variant effects in isolation.

Current statistical methods for GWAS and fine-mapping generally rely on the genotypes of individual variants and ignore their phase. Further, to understand the complexity of a locus when there are multiple associated variants, conditional analysis is frequently implemented to identify whether these variants are independently associated at a locus or if they are all marking the same single polymorphism [7,8]. By statistically conditioning on the genotype of a primary associated variant in a genetic association analysis, variants that remain significant are interpreted as having effects that cannot be explained by the primary variant alone [9–11]. However, we show by simulation that conditional analysis cannot determine whether these effects are independent, such as representing allelic heterogeneity, or coordinated through phase relationships. Moreover, strong linkage disequilibrium (LD) can hinder the discovery of secondary variants. An alternative strategy is to analyze the interaction effects between two variants through epistasis analysis, which typically tests the genotype interaction effects between two variants. This approach often compares a full regression model that includes

both genotype terms and their interactions with a reduced model that excludes the interaction terms [12–14]. Some studies have proposed testing interaction effects based on linkage disequilibrium (LD) [15] or haplotype frequencies [16] in case-control studies to increase statistical power, allowing the incorporation of phase information indirectly. A key limitation of these epistasis tests is their lack of biological interpretability. A significant interaction effect does not clarify whether the effect is *cis* or *trans*, nor whether the variants' effects are truly coordinated. The haplotype association methods [17,18], on the other hand, incorporate phase information under regression frameworks but primarily focus on *cis* rather than *trans* relationships and often do not account for the marginal effects contributed by each locus independently. These limitations hinder a comprehensive analysis of phase effects at GWAS-significant loci with current statistical methodologies.

To address these limitations, we developed the Regression on Phase (RoP) method, which incorporates phase information of variants in a regression framework. For simplicity, this study focuses on analyzing phase effects between two biallelic loci to demonstrate the performance of RoP, though the method can be extended to accommodate other variant types across multiple loci. Analogous to the additive genotype coding at a single locus, we introduce "phasetypes" to represent the effects of *cis* and *trans* relationships additively. RoP tests the phenotypic associations of *cis* and *trans* phasetypes separately while accounting for each variant's main effect. Our simulations show that RoP maintains appropriate type I error (T1E) when variants influence phenotypes independently and outperforms traditional indirect epistasis tests in detecting true phase effects, especially in distinguishing *cis* from *trans* mechanisms.

We analyzed the phase effects at CF genetic modifier loci for lung and intestinal disease severity discovered through GWAS, with phased sequence data generated using the 10XG technology on DNA from individuals with CF enrolled in the Canadian CF Gene Modifier Study [2,19]. GWAS using genotyping arrays have identified several modifier loci, including the *trypsinogen* locus on chromosome 7q35 [19] and the *SLC6A14* locus on chromosome X [19–21]. GWAS identified two association peaks at the *trypsinogen* locus with intestinal obstruction at birth in CF (meconium ileus, MI) in different LD regions. The locus displays complex genetic architecture with five homologous genes, two transcribed (*PRSS1* and *PRSS2*) and three pseudogenes. Fine-mapping with long- and linked-read sequencing revealed a 20 kb common deletion and a missense variant (rs62473563), each in LD with different GWAS association peaks [2], and conditional analysis was implemented to demonstrate evidence of allelic heterogeneity. The *SLC6A14* locus influences both MI and CF-related lung disease but exhibits tissue specificity: variants associated with lung disease map to a distant intergenic enhancer influencing *SLC6A14* expression in airway epithelia, while variants associated with MI lie in a proximal promoter region affecting expression in the pancreas [19] (Fig 1B). Traditional single-variant, genotype-based association analysis cannot determine whether these putatively functional variants act independently or require specific phase configurations to manifest their effects, which is important when designing follow-up functional investigations. We show that the RoP method can leverage the added resolution of long-read sequencing to address this gap.

## Method

### The regression on phase (RoP) approach for testing *cis* and *trans* effects

For two biallelic loci with alleles {A, a} and {B, b}, there are ten distinct diplotypes, corresponding to four *cis* and four *trans* relations. Similar to the genotype of a biallelic locus, we code the phase relationships additively, referred to as phasetypes, which take values in {0, 1, 2}

representing the number of a particular phase relationship carried by an individual's diplo-type, as shown in Table 1.

We assume that either the *cis* relationship or the *trans* relationship of two loci contributes to the outcome. The RoP approach tests phase effects based on a generalized linear model:

$$g(E(Y)) = \beta_0 + \beta_{G_A} \cdot G_A + \beta_{D_A} \cdot D_A + \beta_{G_B} \cdot G_B + \beta_{D_B} \cdot D_B + \beta_{P_{cis}} \cdot P_{cis} + \beta_{P_{trans}} \cdot P_{trans}$$

The phase effects that are in-*cis* or in-*trans* are tested separately with two 1 degree of free-dom (df) tests: $H_0 : \beta_{Pcis} = 0$ and $H_0 : \beta_{Ptrans} = 0$, respectively. $G_A$ and $G_B$ are the additive geno-types, and $D_A$ and $D_B$ are the dominance terms, which equal 1 for heterozygous genotypes and 0 otherwise (Table 1). These terms are included to adjust for the independent effects from each locus. $P_{cis}$ and $P_{trans}$ are phasetypes for *cis* and *trans* relationships. Throughout this study, we let $P_{cis} = Cis_{AB}$ and $P_{trans} = Trans_{AB}$, i.e., the phasetypes coded based on the alternative alleles for the two loci.

The method is invariant to the reference allele selection for $P_{cis}$ and $P_{trans}$; that is, the test statistics have the same value when $P_{cis}$ equals $Cis_{AB}$, $Cis_{Ab}$, $Cis_{aB}$ or $Cis_{ab}$, and when $P_{trans}$ equals $Trans_{AB}$, $Trans_{Ab}$, $Trans_{aB}$ or $Trans_{ab}$. This invariant property is due to the linear rela-tionships between the additive genotypes and phasetypes:

$$Cis_{AB} + Cis_{Ab} = Trans_{AB} + Trans_{Ab} = G_A$$

$$Cis_{AB} + Cis_{aB} = Trans_{AB} + Trans_{aB} = G_B$$

$$\sum_{i\in\{A,a\}} \sum_{j\in\{B,b\}} Cis_{ij} = \sum_{i\in\{A,a\}} \sum_{j\in\{B,b\}} Trans_{ij} = 2$$

and follows from the theorem proven by Chen *et al* for generalized linear models [22]:

**Theorem 1.** *For two generalized linear model $M_1$: $g(E(Y|X)) = \boldsymbol{\beta}_C \cdot X_C + \beta_P^1 \cdot X_P^1$ and $M_2$: $g(E(Y|X)) = \boldsymbol{\beta}_C \cdot X_C + \beta_P^2 \cdot X_P^2$ that have the same link function g. $X_C$ is a $n \times p$ matrix for p secondary covariates or confounders. $X_P^1$ and $X_P^2$ are $n \times q$ matrices for q primary covariates of interest. Then testing the association of primary covariates using Score, LRT or Wald tests is identical under $M_1$ ($H_0 : \beta_P^1 = 0$) and $M_2$ ($H_0 : \beta_P^2 = 0$) if 1) The primary covariates of the two models follows a linear relationship: $X_P^2 = A_C X_C + A_P X_P^1$, where $A_C$ and $A_P$ are $p \times q$ and $q \times q$ transformation matrices; 2) $A_P$ is invertible.*

**Table 1. The additive phasetypes for 10 possible diplotypes of two biallelic loci.** $G_A$ and $G_B$ are additive genotypes. $D_A$ and $D_B$ are dominance terms.

| Diplotypes | Additive | | Dominance | | Additive Phasetypes | | | | | | | |
|---|---|---|---|---|---|---|---|---|---|---|---|---|
| | $G_A$ | $G_B$ | $D_A$ | $D_B$ | $Cis_{AB}$ | $Cis_{aB}$ | $Cis_{Ab}$ | $Cis_{ab}$ | $Trans_{AB}$ | $Trans_{aB}$ | $Trans_{Ab}$ | $Trans_{ab}$ |
| ab/ab | 0 | 0 | 0 | 0 | 0 | 0 | 0 | 2 | 0 | 0 | 0 | 2 |
| Ab/ab | 1 | 0 | 1 | 0 | 0 | 0 | 1 | 1 | 0 | 0 | 1 | 1 |
| aB/ab | 0 | 1 | 0 | 1 | 0 | 1 | 0 | 1 | 0 | 1 | 0 | 1 |
| AB/ab | 1 | 1 | 1 | 1 | 1 | 0 | 0 | 1 | 0 | 1 | 1 | 0 |
| Ab/aB | 1 | 1 | 1 | 1 | 0 | 1 | 1 | 0 | 1 | 0 | 0 | 1 |
| Ab/Ab | 2 | 0 | 0 | 0 | 0 | 0 | 2 | 0 | 0 | 0 | 2 | 0 |
| aB/aB | 0 | 2 | 0 | 0 | 0 | 2 | 0 | 0 | 0 | 2 | 0 | 0 |
| AB/aB | 1 | 2 | 1 | 0 | 1 | 1 | 0 | 0 | 1 | 1 | 0 | 0 |
| AB/Ab | 2 | 1 | 0 | 1 | 1 | 0 | 1 | 0 | 1 | 0 | 1 | 0 |
| AB/AB | 2 | 2 | 0 | 0 | 2 | 0 | 0 | 0 | 2 | 0 | 0 | 0 |

The additive phasetypes satisfy the conditions of the theorem. For example, $Cis_{ab}$ and $Cis_{AB}$ have the relation:

$$Cis_{ab} = \begin{bmatrix} 2 & -1 & -1 & 0 \end{bmatrix} \cdot \begin{bmatrix} 1 \\ G_A \\ G_B \\ Trans_{AB} \end{bmatrix} + 1 \cdot Cis_{AB}$$

Thus, testing for the *cis* effect with $Cis_{AB}$ and $Cis_{ab}$ are equivalent. The same property can be derived between other *cis* relationship pairs and between *trans* relationships (S1 Text).

When the reference alleles are correctly specified for the phasetypes, the regression coefficients provide unbiased estimates of phase effects, which is the expected change in phenotype per additional count of the phase relationship carried by the homologous chromosomes. However, if the reference alleles are misspecified, the coefficients retain the correct magnitude but may have an opposite direction relative to the true phase effects. For example, when the true phase effect is driven by $Cis_{AB}$ or $Trans_{AB}$, estimates for $P_{Cis} = Cis_{ab}$ and $P_{trans} = Trans_{ab}$ remain unbiased, whereas those for $Cis_{aB}$, $Trans_{aB}$, $Cis_{Ab}$, and $Trans_{Ab}$ have the same magnitude but opposite in sign (S2 text). Since the true risk alleles are typically unknown, the RoP tests focus on detecting the presence of phase effects rather than determining the direction of the effects. Importantly, the predicted phenotype remains unchanged regardless of the choice of reference alleles (S2 Text).

The proposed method can be extended to accommodate multi-allelic variants. For multi-allelic variant A with m alleles $\{A_1, A_2, ..., A_m\}$ and variant B with n alleles $\{B_1, B_2, ..., B_n\}$, the RoP model is formulated as:

$$g(E(Y)) = \beta_0 + \sum_{i=2}^{m} \beta_{G_{A_i}} G_{A_i} + \sum_{i=2}^{m} \beta_{D_{A_i}} D_{A_i} + \sum_{j=2}^{n} \beta_{G_{B_j}} G_{B_j} + \sum_{j=2}^{n} \beta_{D_{B_j}} D_{B_j}$$
$$+ \sum_{i=2}^{m}\sum_{j=2}^{n} \beta_{ij}^{cis} Cis_{A_iB_j} + \sum_{i=2}^{m}\sum_{j=2}^{n} \beta_{ij}^{trans} Trans_{A_iB_j}$$

where $G_{A_i}$, $G_{B_j}$, $D_{A_i}$ and $D_{B_j}$ are the additive and dominance genotypes for allele $A_i$ and $B_j$, respectively, and $Cis_{A_iB_j}$ and $Trans_{A_iB_j}$ are the corresponding additive phasetypes. The *cis* and *trans* effects between the two variants can be tested by

$$H_0 : \beta_{A_2B_2}^{cis} = \beta_{A_2B_3}^{cis} = \cdots = \beta_{A_mB_n}^{cis} = 0 \quad \text{and} \quad H_0 : \beta_{A_2B_2}^{trans} = \beta_{A_2B_3}^{trans} = \cdots = \beta_{A_mB_n}^{trans} = 0$$

each takes $(m-1)(n-1)$ df. The genotypes and phasetypes for the reference alleles $\{A_1, B_1\}$ can be omitted due to the linear dependencies between the additive genotypes and phasetypes. This formulation allows RoP to handle LD blocks by treating each distinct haplotype within a block as an allele in the multi-allelic model, which facilitates its application in genome-wide analysis.

## Methods for epistasis analysis and haplotype regression

We compared the performance of our Regression on Phase (RoP) method against traditional epistasis tests, which focus on detecting interaction effects between two specific genetic variants while controlling for marginal effects. We focused on a set of benchmark approaches that are commonly used and align with the objectives of our study. Methods designed for exhaustive searches of epistasis signals across the entire genome or large genomic regions are not considered here [23,24].

The most widely employed strategy for testing epistasis effects between two variants is to include genotype interactions in a regression framework [13,14]. Using this approach, each variant's genotype is modelled additively, and an interaction term tests for deviation from additivity:

$$g(E(Y)) = \beta_0 + \beta_{G_A} \cdot G_A + \beta_{G_B} \cdot G_B + \beta_{G_A G_B} \cdot G_A \cdot G_B$$

where the presence of an epistasis effect is assessed through a 1 df test of $H_0 : \beta_{G_A G_B} = 0$.

Some extensions incorporate dominance terms:

$$\begin{aligned} g(E(Y)) = {} & \beta_0 + \beta_{G_A} \cdot G_A + \beta_{D_A} \cdot D_A + \beta_{G_B} \cdot G_B + \beta_{D_B} \cdot D_B \\ & + \beta_{G_A G_B} \cdot G_A \cdot G_B + \beta_{G_A D_B} \cdot G_A \cdot D_B + \beta_{D_A G_B} \cdot D_A \cdot G_B \\ & + \beta_{D_A D_B} \cdot D_A \cdot D_B \end{aligned}$$

where epistasis is evaluated using a 4 df test $H_0 : \beta_{G_A G_B} = \beta_{G_A D_B} = \beta_{D_A G_B} = \beta_{D_A D_B} = 0$.

When phase information is available, Schaid [18] proposed a saturated model that includes an additional phase term:

$$\begin{aligned} g(E(Y)) = {} & \beta_0 + \beta_{G_A} \cdot G_A + \beta_{D_A} \cdot D_A + \beta_{G_B} \cdot G_B + \beta_{D_B} \cdot D_B \\ & + \beta_{G_A G_B} \cdot G_A \cdot G_B + \beta_{G_A D_B} \cdot G_A \cdot D_B + \beta_{D_A G_B} \cdot D_A \cdot G_B \\ & + \beta_{D_A D_B} \cdot D_A \cdot D_B + \beta_V \cdot V \end{aligned}$$

the phase term $V = 1$ when both loci are heterozygous and the alternative alleles are in-*cis* (i.e. $G_A = G_B = Cis_{AB} = 1$) and $V = 0$ otherwise. Both epistasis and phase effects can be evaluated simultaneously using a 5 df test $H_0 : \beta_{G_A G_B} = \beta_{G_A D_B} = \beta_{D_A G_B} = \beta_{D_A D_B} = \beta_V = 0$.

In addition to regression-based tests, other methods have been proposed that incorporate haplotype frequencies to improve power in case-control studies. Among these, we consider the haplotype odds ratio (OR) test [16] as a benchmark. This approach leverages phase information, albeit indirectly through haplotype frequencies, making it a relevant comparator for our RoP method, which directly models phase effects.

Specifically, the haplotype frequencies $f_*^1$ for cases and $f_*^0$ for controls are used to define the log-odds ratio:

$$log(OR) = log\left(\frac{f_{AB}^1 f_{ab}^1}{f_{Ab}^1 f_{aB}^1}\right) - log\left(\frac{f_{AB}^0 f_{ab}^0}{f_{Ab}^0 f_{aB}^0}\right)$$

The OR test tests for the same null hypothesis as the 1 df interaction test, $H_0 : \beta_{G_A G_B} = 0$, using the test statistic:

$$\chi^2 = \frac{log(\hat{OR})^2}{\hat{v}^1 + \hat{v}^0}$$

where

$$\hat{v}^i = \frac{1}{2n^i}\left(\frac{1}{\hat{f}_{AB}^i} + \frac{1}{\hat{f}_{aB}^i} + \frac{1}{\hat{f}_{Ab}^i} + \frac{1}{\hat{f}_{AB}^i}\right)$$

The resulting test statistic follows a $\chi^2$ distribution with 1 df under the null hypothesis. Notably, under the rare disease assumption, $log(OR)$ approximates the genotype interaction effect $\beta_{G_A G_B}$ [25], and as we show later, it also approximates *cis* effects $\beta_{cis}$.

When phase information is available, a natural alternative to RoP is haplotype regression, in which each haplotype of two biallelic variants is treated as an allele of a single multi-allelic

marker and modelled additively in a generalized linear model. This approach is equivalent to jointly testing the effect of the four *cis* relationships without adjusting for the marginal effects:

$$g(E(Y)) = \beta_0 + \beta_{AB} \cdot Cis_{AB} + \beta_{Ab} \cdot Cis_{Ab} + \beta_{aB} \cdot Cis_{aB}$$

and the effects of haplotype are evaluated by the 3 df test: $H_0 : \beta_{AB} = \beta_{Ab} = \beta_{aB} = 0$. $Cis_{ab}$ is used as the reference.

This approach has two key limitations for detecting coordinated phase effects. First, it is sensitive to the marginal effects, and a significant result cannot reveal whether the signal arises from independent main effects or from their coordinated phase (S1A Fig). Second, because only the four *cis* relationships are modelled, the method cannot identify which phase configuration drives the association, and it is underpowered in detecting *trans* effects (S1B and S1C Fig). Due to these limitations, haplotype regression was not included as a benchmark in our simulations.

## Phase analysis at cystic fibrosis modifier loci

We obtained phased whole genome sequence from 564 individuals with CF from the Canadian CF Gene Modifier Study (CGMS) using 10XG linked-read technology, with the intent of investigating the complex variation at the *trypsinogen* and *SLC6A14* CF modifier loci. Details for the sample processing and variant calling are described in [2]. For the association analysis of MI at the *trypsinogen* locus with phase effects between the 20 kb deletion polymorphism and the missense variant, we focused on the same n = 307 individuals as in [2]. At the *SLC6A14* locus, we analyzed the phase effects of the MI-associated promoter variant and the lung disease-associated enhancer variant on gene expression in CF human nasal epithelial (HNE) cells (n = 79), on the age at the first *Pseudomonas aeruginosa* (PsA) infection (n = 41), and on lung function (n = 413) from individuals enrolled in the CGMS. Gene expression of *SLC6A14* from naive HNE cells was obtained by RNA sequencing [26,27], and expression levels were measured using transcript per million counts (TPM). The RNA integrity number (RIN) was included in the regression model to account for variation in RNA quality. The age at the first PsA infection was defined as in [28] and [29]. The residuals after regressing age at infection on the current calendar age were used as the outcome for the association analysis [29]. Lung disease severity was measured by SakNorm as in the CF lung GWAS study [20], which was calculated by the quantile of averaged forced expiratory volume in 1 second (FEV1) measurements over a three-year period, adjusted for age, height, sex, and CF-specific survival rate [30]. The quantiles were generated using lung function reference equations based on the CGMS cohort [31].

## Results

### Limitations of conditional analysis to distinguish allelic heterogeneity from phase effects

We evaluated the ability of conditional analysis to distinguish between allelic heterogeneity and phase effects through a simulation study. Haplotypes for two biallelic loci were generated under Hardy-Weinberg equilibrium, considering two LD scenarios (D' = 0 and D' = 0.8). The frequency of the primary variant ($P_A$) was fixed at 0.2, while the frequency of the secondary variant ($P_B$) was varied from 0.05 to 0.95. A continuous outcome was simulated where the loci had independent effects and where their effects were coordinated through their *cis* or *trans* relationships (Table 2).

**Table 2. Models for simulations.** Continuous outcomes are used to assess power in conditional analysis, and binary outcomes are used to compare T1E and the power of RoP with epistasis tests. For logistic models, $\beta_0 = -2$ corresponds to a baseline prevalence of $P(Y = 1) = 0.12$. $I$ are indicator variables corresponding to dominant and recessive inheritance patterns. For T1E, we set $\beta_{G_A} = \beta_{G_B} = 1$. For power, $\beta_P = 0.5$ for additive and dominant models and $\beta_P = 1.5$ for the recessive *cis* effect.

| Power of conditional analysis | |
|---|---|
| Model 1: Allelic heterogeneity | $Y = -2 + 0.5 \cdot G_A + 0.5 \cdot G_B + \epsilon, \epsilon \sim N(0,1)$ |
| Model 2: Additive *cis* effect | $Y = -2 + 1.5 \cdot P_{cis} + \epsilon, \epsilon \sim N(0,1)$ |
| Model 3: Additive *trans* effect | $Y = -2 + 1.5 \cdot P_{trans} + \epsilon, \epsilon \sim N(0,1)$ |
| **Type I Error and Power** | |
| *Type I error* | |
| Model 1: Additive effect on locus A | $logit(p) = -2 + 1 \cdot G_A$ |
| Model 2: Dominant effect on locus A | $logit(p) = -2 + 1 \cdot I_{G_A \geq 1}$ |
| Model 3: Recessive effect on locus A | $logit(p) = -2 + 1 \cdot I_{G_A = 2}$ |
| Model 4: Additive effects on both loci | $logit(p) = -2 + 1 \cdot G_A + 1 \cdot G_B$ |
| Model 5: Dominant effects on both loci | $logit(p) = -2 + 1 \cdot I_{G_A \geq 1} + 1 \cdot I_{G_B \geq 1}$ |
| Model 6: Recessive effects on both loci | $logit(p) = -2 + 1 \cdot I_{G_A = 2} + 1 \cdot I_{G_B = 2}$ |
| *Power* | |
| Model 1: Additive *cis* effect | $logit(p) = -2 + 0.5 \cdot P_{cis}$ |
| Model 2: Dominant *cis* effect | $logit(p) = -2 + 0.5 \cdot I_{P_{cis} \geq 1}$ |
| Model 3: Recessive *cis* effect | $logit(p) = -2 + 1.5 \cdot I_{P_{cis} = 2}$ |
| Model 4: Additive *trans* effect | $logit(p) = -2 + 0.5 \cdot P_{trans}$ |
| Model 5: Dominant *trans* effect | $logit(p) = -2 + 0.5 \cdot I_{P_{trans} \geq 1}$ |

Conditional analysis alone is insufficient to reliably distinguish between allelic heterogeneity and phase effects. The power to detect the secondary variant (SNP B) when conditioning on the primary variant (SNP A) is primarily influenced by the LD between the two variants, regardless of whether their effects were independent or coordinated through their phase (Fig 2). When the two variants were not in LD, conditioning on the primary SNP increased the power of detecting the secondary SNP. In contrast, when the two variants were in LD, the conditional analysis reduced the power of detecting the secondary variant, especially in the presence of phase effects. This indicates that conditional analysis may fail to identify variants that exert phase effects within a high LD region. The RoP method explicitly models *cis* and *trans* configurations while adjusting for marginal effects. As a result, it is sensitive to phase effects (Fig 2B, 2C, 2E, and 2F) but not to allelic heterogeneity (Fig 2A and 2D), enabling the method to distinguish these two mechanisms. Other non-conditional methods, such as epistasis tests, also adjust for marginal effects and detect the coordinated interactions. However, as demonstrated by simulation results in the following section, these methods cannot determine which phase configuration drives the association signal.

## RoP outperforms epistasis tests in detecting phase effects

To evaluate and compare methods suitable for detecting phase effects between two selected variants, we conducted simulations to assess T1E and power for the RoP method and traditional epistasis tests. Haplotypes for two biallelic loci were simulated as in the previous section. While the RoP method can be applied to quantitative outcomes, we simulated binary traits to compare its performance with the haplotype OR test. Case-control outcomes were generated by logistic models with 1,000 unrelated individuals per group, following the scenarios detailed in Table 2. To assess T1E, we examined six configurations in which independent effects originated from one or both loci under additive, dominant, and recessive inheritance patterns. For power analysis, we modelled additive and dominant phase effects for both *cis*

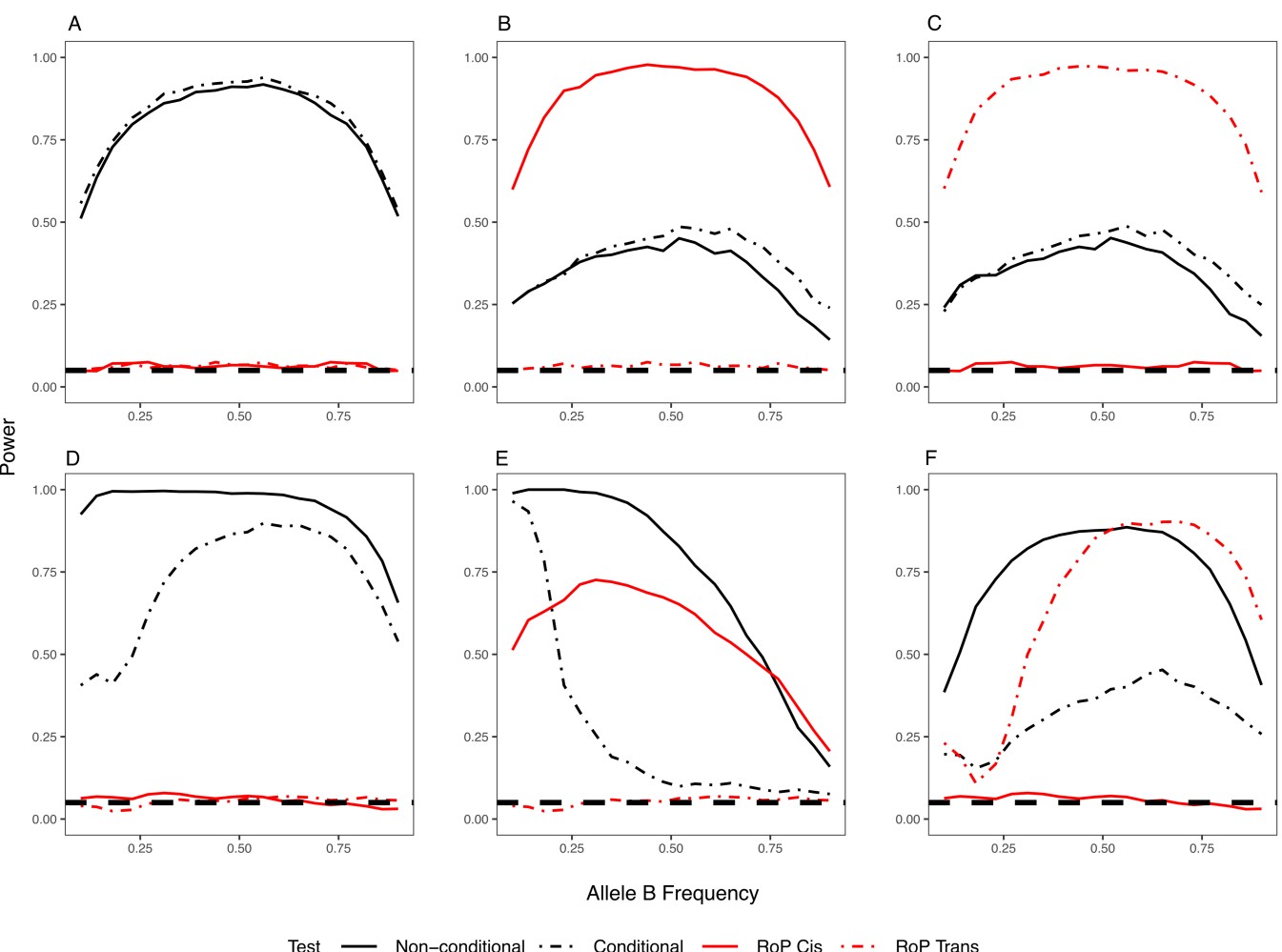

**Fig 2. The power of detecting the secondary SNP B with (conditional) and without (non-conditional) adjusting for the genotype of primary SNP A, compared with the power of RoP in detecting *cis* and *trans* effects.** From left to right: (A) (D) allelic heterogeneity, (B) (E) *cis* and (C) (F) *trans* effects; (A)-(C): the variants are not in LD (D' = 0) and (D)-(F) in LD (D' = 0.8). A reference line at T1E = 0.05 was added to illustrate that RoP maintains nominal type I error under allelic heterogeneity.

and *trans* relationships, and recessive phase effects for *cis* relationships only. Both T1E and power were calculated over 1,000 iterations.

The 1 df genotype interaction test and the haplotype OR test exhibited inflated T1E in the presence of independent effects (Figs 3 and S2 and S1 Table). Specifically, the 1 df interaction test showed inflated T1E when the two loci were in LD and one of them carried either dominant or recessive marginal effects. The OR test maintained T1E control when only one locus had a marginal effect (S2 Fig), but inflation occurred if both loci had marginal effects, irrespective of their LD relationship (Fig 3). This is consistent with previous analytical results, which demonstrated that the OR test is only invariant to marginal effects contributed by a single locus [25]. In contrast, the RoP test, the saturated test, and the 4 df genotype interaction test maintained correct T1E across all scenarios, demonstrating their robustness in distinguishing true phase effects from independent contributions of individual variants.

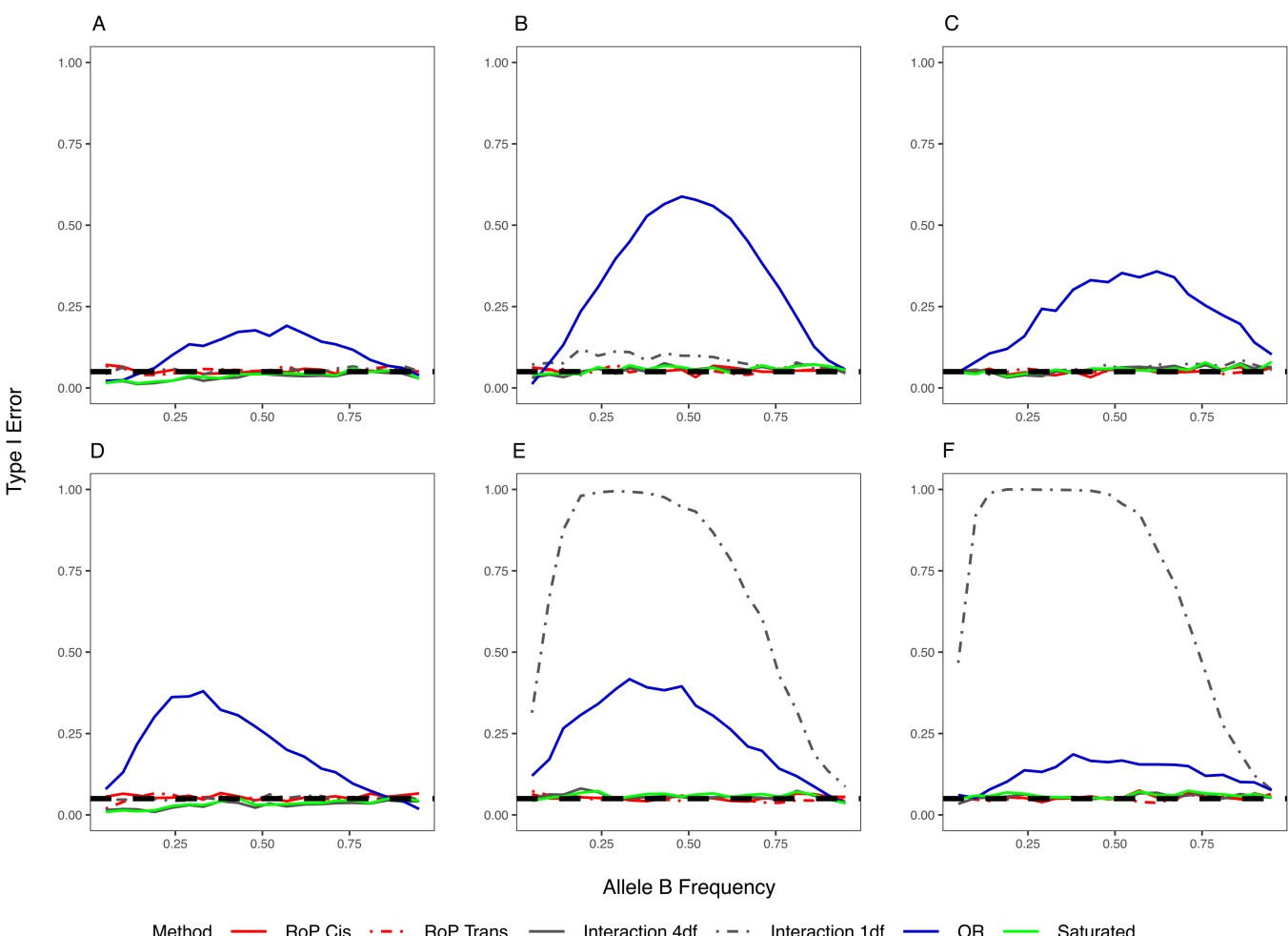

**Fig 3. The type I error when independent effects occurred at both loci.** Additive (A)(D), dominant (B)(E) and recessive (C)(F) inheritance models were examined; (A)-(C): the variants are not in LD (D' = 0) and (D)-(F): in LD (D' = 0.8). Dashed lines correspond to a type I error of 0.05.

Fig 4 presents the power to detect additive phase effects. The RoP tests outperformed other methods and effectively distinguished between *cis* and *trans* effects: when a *cis* relationship between two variants contributed to the outcome, the power to detect the *trans* effect was controlled at a level near 0.05, and vice versa. Similar results were also observed for dominant phase effects (S3A, S3C, S3D, and S3F Fig). However, the RoP tests showed reduced power in detecting recessive *cis* effects, whereas the 4 df genotype interaction test exhibited superior performance (S3B and S3E Fig). The haplotype OR test, which incorporates phase information indirectly, demonstrated identical power to the RoP test in detecting additive and dominant *cis* effects but was ineffective for detecting *trans* effects. Analytical findings (S3 Text) confirm that the log haplotype frequency OR approximates additive *cis* effects but equals zero under *trans* effects, explaining its lack of power in *trans* scenarios. The saturated test, which incorporates an additional phase term, outperforms the 4 df genotype interaction test in detecting additive and dominant phase effects but exhibits lower power compared to RoP. In fact, the additive *cis* and *trans* effects can be written as linear combinations of the four interaction terms plus the phase term (S4 Text), indicating that the saturated model does

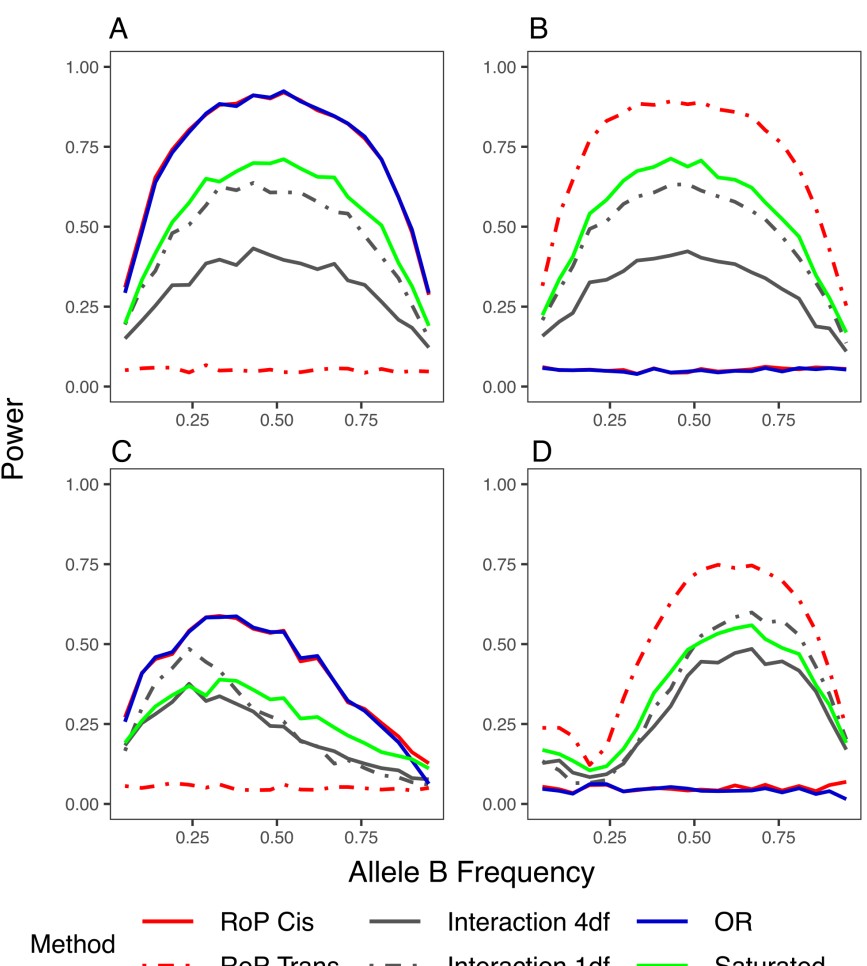

**Fig 4. The power of detecting additive phase effects that are in *cis* (A)(C) and in *trans* (B)(D).** (A) and (B): the variants are not in LD (D' = 0); (C) and (D): the variants are in LD (D' = 0.8). Note, the OR test (blue) and 1 df genotype interaction test (grey dashed) have inflated T1E when there is a main effect, as in Fig 3.

capture these effects, but the additional degree of freedom reduces its power relative to the RoP framework. The saturated test cannot distinguish the *cis* and *trans* mechanisms, as the direction of the phase term coefficient $\beta_V$ relies on both the selection of reference alleles for phasetypes and the contribution of *cis* or *trans* relationships (S4 Text).

While the RoP method performed strongly for additive and dominant phase effects, detecting recessive *cis* effects remains challenging. This mirrors the findings from single-locus association studies, where the additive genotype coding often lacks power for recessive variants [32,33]. Zhou and Dizier [34,35] showed that the power can be improved by jointly testing for additive and dominance terms. To enhance the RoP approach under recessive inheritance, we introduced the dominance phase terms $D_{cis}$ and $D_{trans}$, which equal 1 when $P_{cis}$ and $P_{trans}$ are 1, respectively. We then tested recessive *cis* effects using $H_0 : \beta_{P_{cis}} = \beta_{D_{cis}} = 0$. Contrary to the expectations based on single-locus results, this adjustment reduces the power of recessive *cis* effect detection (Fig 5 Model 1). This is because the phenotypic variation attributed to the recessive *cis* effect can be effectively explained by the additive and dominance *trans* terms in the model (S4 Text). Removing *trans* terms can improve the power for detecting recessive

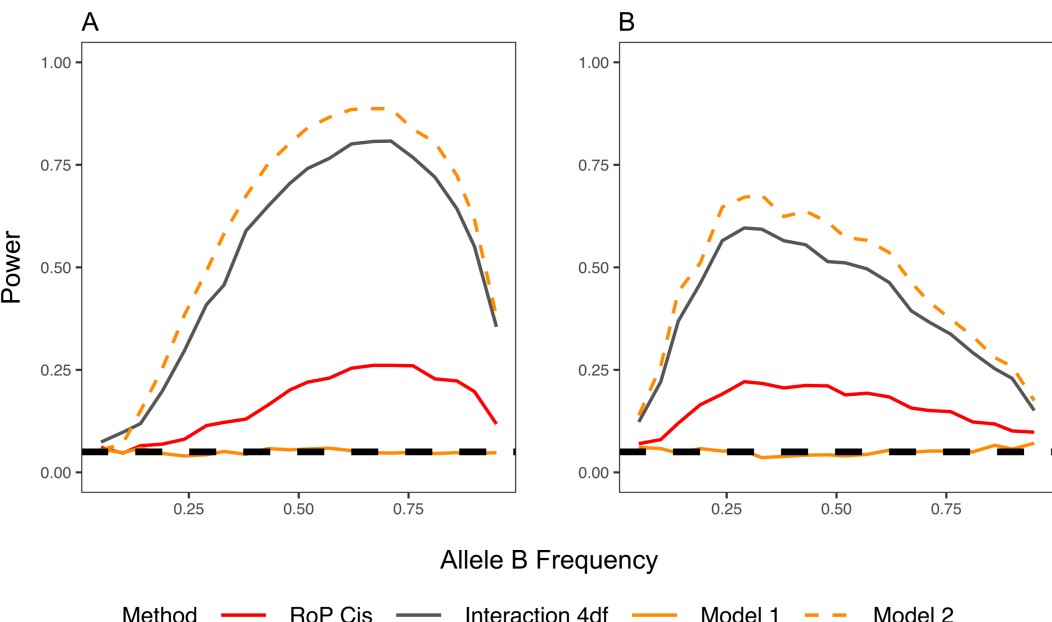

**Fig 5. Power to detect recessive *cis* effects.** (A) D' = 0; (B) D' = 0.8. Model 1: $g(E(Y))) = \mathbf{G} \cdot \beta_{\mathbf{G}} + \beta_{P_{cis}} \cdot P_{cis} + \beta_{D_{cis}} \cdot D_{cis} + \beta_{P_{trans}} \cdot P_{trans} + \beta_{D_{trans}} \cdot D_{trans}$; Model 2: $g(E(Y))) = \mathbf{G} \cdot \beta_{\mathbf{G}} + \beta_{P_{cis}} \cdot P_{cis} + \beta_{D_{cis}} \cdot D_{cis}$. Recessive *cis* effects were tested by $H_0 : \beta_{P_{cis}} = \beta_{D_{cis}} = 0$.

*cis* effects (Fig 5 Model 2) but risk T1E inflation if the true effect is in-*trans*. The 4 df genotype interaction test demonstrated the best power to detect the recessive *cis* effect. This is because the recessive *cis* model can be written as the linear combination of the interactions between additive and dominance genotype terms, even though these terms did not incorporate phase information (S4 Text). The saturated test exhibited slightly lower power than the 4 df interaction test as a result of the extra 1 df for the phase term, which does not contribute to detecting recessive *cis* effects. Based on these observed tradeoffs, when the RoP method is not significant for both *cis* and *trans* effects, we suggest applying the 4 df genotype interaction test to assess the possibility of a recessive *cis* relationship with the phenotype.

LD between variants can influence the power of RoP tests. Overall, LD reduces power for both *cis* and *trans* effect detection, with a more pronounced effect on *trans* effects when the two variants share similar allele frequencies. To further elucidate how LD influences the ability of the RoP method to detect phase effects at different allele frequencies, we performed detailed numerical analyses under a linear regression framework (Fig 6). We modelled a quantitative outcome with a sample size of n = 1,000, residual standard error $\sigma = 2$, and coefficient $\beta_P = 1$ for additive phase effects. Power was evaluated at varying allele frequencies and compared between scenarios with no LD (D'= 0) and strong LD (D' = 0.8). For *cis* effects (Fig 6A–6C), LD can either enhance or reduce power depending on allele frequencies. When both loci are rare, LD increases the power to detect *cis* effects. In contrast, when one allele is rare and the other common, LD introduces a stronger correlation between the *cis* term and the rare variant's genotype, thus diminishing power. Under complete LD (D' = 1), the *cis* term for a rare-common pair essentially became indistinguishable from the genotype of the rare variant, making it impossible to separate *cis* effects from marginal effects. The *trans* effect detection showed greater power reduction when allele frequencies were similar, regardless of whether the variants were rare or common, and it was less affected when allele frequencies

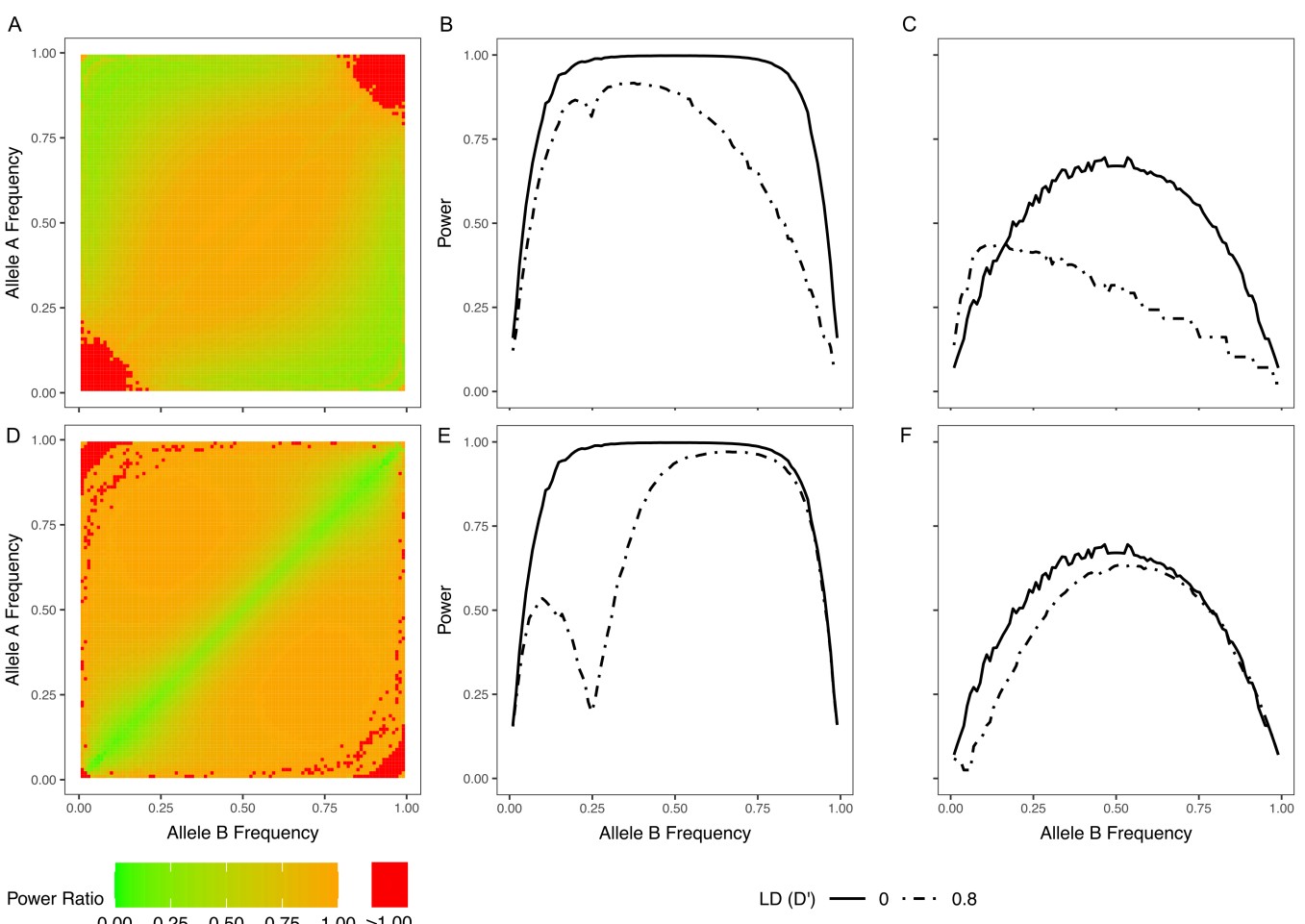

**Fig 6. The impact of LD on RoP.** (A) and (D): The ratio of the power when D' = 0.8 over the power when D' = 0 to detect *cis* and *trans* effects, respectively. Green areas correspond to scenarios where LD reduces power, and red areas correspond to improved power. (B) and (C): the power to detect *cis* effects when A allele is common ($P_A = 0.25$) and rare ($P_A = 0.05$); (E) and (F): same as (B) and (C) but for *trans* effects.

were different (Figs 4B, 4D, and 6D–6F). This is because the three diplotypes (Ab/Ab, aB/aB, Ab/aB) became infrequent when two variants had similar frequencies and are in LD, inducing stronger multicollinearity between the *trans* term and the main effect terms in the remaining diplotypes. Given these complexities, especially in smaller samples or when the underlying mechanism of phase effects is unknown, focusing on variant pairs that are not in strong LD may be more efficient.

### Investigating phase effects at cystic fibrosis modifier loci: Allelic heterogeneity and tissue specificity

We used the 10XG linked-read data from the CGMS cohort to investigate whether GWAS-identified variants at the *trypsinogen* and *SLC6A14* loci contribute to CF disease severity independently or through coordinated phase effects.

At the *trypsinogen* locus, the missense variant rs62473563 and the 20 kb common deletion, which reside in different LD blocks, are associated with MI and serve as eQTLs for *PRSS2* expression in the pancreas. Conditional analysis suggested that accounting for one variant

reduced the p-value for the other's association with MI, leading to the conclusion that the two variants contribute to MI risk independently [2]. Our simulation study showed that the changes in p-value in the conditional analysis are strongly influenced by LD and cannot disentangle allelic heterogeneity and phase effects. Instead, we evaluated phase effects between the deletion and the missense variant at the *trypsinogen* locus by the RoP method. The *cis* configuration involving the deletion and rs62473563 T allele did not occur in our dataset, and the remaining *cis* terms were linearly dependent on genotypes. Consequently, we focused on *trans* effects using the RoP method, and complementary analyses using the haplotype odds ratio and the 4 df genotype interaction tests for *cis* effects. None of these tests revealed significant associations with MI risk ($p_{trans} = 0.95$, $p_{OR} = 0.83$, $p_{Interaction} = 0.95$), supporting the original conclusion that these two variants independently influence MI risk rather than acting through a phase-dependent mechanism.

The *SLC6A14* locus displayed tissue-specific associations in previous CF GWASs (Fig 1B). A cluster of variants located in an intergenic region approximately 200 kb upstream of the *SLC6A14* transcription start site (TSS), annotated closer to *AGTR2* than *SLC6A14*, were GWAS-significantly associated with CF lung function. One of the associated variants in the cluster, rs4446858, disrupts an IRF1/IRF2 binding site [36] annotated within a region with a distal enhancer-like signature (E2765449/enhD) [37–39], indicating possible immune-related regulatory mechanisms. The top MI-associated SNP, rs3788766, which resides ∼2.4 kb upstream of *SLC6A14* TSS, showed no individual effect on lung disease (Figure 3 in [19]). To evaluate the potential role of the rs3788766 *SLC6A14* promoter variant in CF lung disease, we analyzed the phase effects of rs3788766 and rs4446858 on CF lung disease severity (SakNorm), the age at first PsA infection and the expression level of *SLC6A14* in CF HNE (Table 3). Analyses were stratified by sex to address the impact of X chromosome inactivation [40,41]. Replicating GWAS findings with our 10XG data, we confirmed that rs4446858 alone, but not rs3788766, was associated with *SLC6A14* expression and CF lung function in females. Neither variant alone was significantly associated with the age at first PsA infection at the 0.05 level. However, when analyzing their phase relationship, we found that their *cis* configuration significantly affected *SLC6A14* expression and the age at first PsA infection in both males and females, whereas the *trans* configuration showed no significant effect.

**Table 3. The association p-values for the *SLC6A14* locus with lung phenotypes.** The *trans* effects were tested only in females. The age at 1st PsA infection phenotype was measured by the residual of the age at the first infection regressed on the calendar age [29]. Lung disease severity was measured by SakNorm [20,30].

| | *SLC6A14* Expression | Age at 1st PsA Infection | SakNorm |
|---|---|---|---|
| Male | n = 40 | n = 14 | n = 213 |
| rs4446858 | 0.86 | 0.7 | 0.2 |
| rs3788766 | 0.93 | 0.77 | 0.87 |
| RoP *Cis* | **0.05** | **0.03** | 0.64 |
| RoP *Trans* | NA | NA | NA |
| Female | n = 39 | n = 27 | n = 200 |
| rs4446858 | **0.05** | 0.89 | **0.005** |
| rs3788766 | 0.98 | 0.26 | 0.13 |
| RoP *Cis* | **0.03** | **0.04** | 0.61 |
| RoP *Trans* | 0.38 | 0.95 | 0.77 |

## Discussion

Long-read sequencing (LRS) will increasingly be adopted in population and disease cohorts for genetic association studies, offering enhanced resolution across extended genetic regions, particularly regarding complex variation and phase information. This advancement enables the investigation of how phase relationships between genetic variants influence traits, which is, in general, overlooked in traditional genetic association analyses. In this study, we assessed the limitations of traditional epistasis methods in detecting phase effects and introduced a new regression-based method, Regression on Phase (RoP). To our knowledge, RoP is the first statistical method developed specifically to analyze *cis* and *trans* phase effects in genetic association studies using LRS data, where the phase is observed over long genomic distances. The method can accommodate both biallelic and multi-allelic variants. Simulations indicate that the RoP method maintains appropriate Type I Error (T1E) rate control when genetic variants contribute independently to the phenotype and outperforms traditional epistasis tests for detecting phase effects, particularly under additive and dominant inheritance patterns. Most importantly, the RoP method effectively distinguishes *cis* and *trans* effects, providing deeper insights into the coordinated action of GWAS-associated variants and the loci that demonstrate evidence of allelic heterogeneity. The method is implemented in the R package *RegPhase* (see Code and Data Availability).

Building on this capability, RoP offers a distinct advantage over the traditional conditional analysis when probing the mechanisms underlying GWAS-significant loci that exhibit multiple association signals, especially when prioritizing variants with functional annotations that are likely to represent causal mechanisms. While conditional analysis is ubiquitous in the field for discovering additional associations that are independent of the primary signal, our simulation study revealed that it cannot reliably distinguish between allelic heterogeneity effects and phase effects. Instead, for two contributing variants, the power to detect the secondary signal under conditional testing is predominantly driven by their linkage, regardless of whether their effects are actually independent or coordinated through phase. Therefore, the persistence of a conditional signal does not necessarily imply allelic heterogeneity but could also reflect phase effects, and conditional analysis alone cannot disentangle these two mechanisms. RoP resolves this ambiguity by explicitly modelling *cis* and *trans* configurations while accounting for marginal genotype effects: RoP detects a significant *cis* or *trans* effect when the two variants interact through their phase configurations, and remains non-significant when the two variants act independently. This result underscores that directly analyzing phase information is essential for resolving the true nature of secondary signals, irrespective of the magnitude of LD, and thus, RoP provides a more comprehensive framework for interpreting the genetic architecture of complex traits.

LD plays a critical role in determining the power of the RoP test. Generally, LD reduces the ability to detect both *cis* and *trans* effects, suggesting that phase effect analyses would benefit from focusing on variants located in different LD regions, especially in studies with limited sample sizes. However, our results suggest the impact of LD can be remedied under certain configurations. In particular, pairs of rare variants in LD may actually improve the detection of *cis* effects, and power losses for *trans* effects can be mitigated by focusing on variants with different allele frequencies. Therefore, the RoP tests can be applied strategically to functionally relevant variants located in the same LD region to maximize the utility of this method for analyzing phase effects.

Although RoP was developed for LRS datasets with accurate phase information, our simulations indicate that it is robust to moderate phasing error and can therefore be applied directly to short-read data where phase is inferred with uncertainty. We simulated haplotypes

for n = 5,000 individuals with two independent variants under three minor allele frequency (MAF) scenarios: 1) both alleles are rare (MAF = 0.05); 2) one rare allele and one common allele (MAF = 0.05 and 0.2, respectively); and 3) both alleles are common (MAF = 0.2). A continuous outcome Y was generated from their additive *cis* relationships, with effect size and residual variance chosen to yield a heritability of $h^2 = 0.003$. Switch errors were introduced only in individuals who are heterozygous at both variants, at rates ranging from 5% to 20%, with 0% (no switch error) serving as the benchmark (S4 Fig and S2 Table). At a 10% switch error rate, RoP retained over 95% of its original power to detect the *cis* effect (relative to no switch error) and maintained nominal T1E for *trans* effects in all MAF scenarios, effectively distinguishing between the two mechanisms. Substantial inflation of T1E was observed at a 20% switch error rate. Most modern phasing tools, such as SHAPEIT5, achieve switch error rates below 5% even for extremely rare variants [42], which is well below the error level at which RoP performance began to degrade in our simulations. These results demonstrate that under realistic short-read phasing accuracy with contemporary methods [42,43], the statistical validity of RoP is preserved.

Our analysis of phase effects at CF disease modifier loci demonstrates the added information obtained from genetic association studies with LRS data over single-variant GWAS, highlighting how phase analysis can uncover otherwise hidden genetic mechanisms. The chr7q35 *trypsinogen* locus includes two protein-coding genes, *PRSS1* and *PRSS2*, which are expressed exclusively in the pancreas. Although our simulation study demonstrated the limitations of the widely implemented conditional analysis, our phase analysis confirmed the allelic heterogeneity at this locus inferred by this technique. These results suggest the missense variant and the common deletion contribute to MI susceptibility independently, potentially involving transcriptional regulation and protein-level effects. Future functional studies should aim to delineate these distinct roles and their downstream biological effects on pancreatic phenotypes and MI. At the *SLC6A14* locus, GWAS results suggested independent associations for lung function and MI, separated by 200 kb, leading to uncertainty in whether *SLC6A14* was the gene contributing to both phenotypes. eQTL analysis indicated both SNPs contributed to *SLC6A14* gene expression variation and revealed tissue-specific effects [19]: the GWAS-significant enhancer variant rs4446858 was associated with CF lung disease severity and *SLC6A14* gene expression in CF HNE but not MI, whereas the promoter variant rs3788766 was associated with MI and *SLC6A14* gene expression in the pancreas but not with respiratory phenotypes, including expression in respiratory models. Our phase analysis revealed that these variants do not operate independently for CF respiratory phenotypes; instead, their *cis* configuration showed significant association with *SLC6A14* airway expression and the age at first PsA infection in the lungs, indicating the promoter variant can influence lung outcomes when acting in tandem with the IRF1/IRF2 binding site variant through gene regulation. This is consistent with known genome biology that enhancers convey regulatory function through their interactions with promoters [44]. Functional investigation by our group supports this *SLC6A14 cis* finding [45]. Reporter gene expression assays confirmed that the genomic region containing rs3788766 functions as a promoter in both pancreatic and lung-derived cell lines, and haplotypes involving rs3788766 altered the reporter gene expression level, establishing its role as a functional promoter variant in both tissues. The region containing rs4446858 alone did not display promoter activity in either tissue; however, when placed adjacent to the promoter, it acted as an enhancer specifically in lung-derived cells. Notably, rs4446858 modulated the enhancer function with the addition of exogenous IRF1, underscoring its immune-related regulatory function in the lung, which aligns with the role of bacterial

infections in CF lung disease [20,26,46]. Although no significant phase effects on lung function (SakNorm) were detected, previous studies have established the association and causal relationships between PsA infection and reduced lung function in CF [29,47,48].

The RoP method can be applied to both GWAS-significant loci and genome-wide analysis. For GWAS-significant loci with multiple association signals, RoP can be used to assess the phase effects between the lead SNPs of the primary and secondary signals, or between variants with potential functional relevance. In such targeted analyses, only one variant pair is tested for each locus, and multiple-testing correction is therefore unnecessary. However, if multiple variant pairs are examined, appropriate multiple-testing adjustments should be applied. For genome-wide analysis, one straightforward approach is to implement a moving window that exhaustively tests phase effects for all variant pairs within fixed genomic regions. However, this strategy may lead to a heavy multiple-testing burden and substantial computational cost. A more efficient alternative aggregates variants in strong LD into haplotypes and tests phase effects between adjacent LD blocks, treating each block as a multi-allelic variant. This approach avoids underpowered tests between highly correlated variants, reduces the multiple-testing burden, and improves efficiency. Further power gains can be achieved by merging similar haplotypes within each LD block [17] prior to analysis, thereby reducing the number of alleles and lowering the degrees of freedom of the test. Because a single LD block can be paired with multiple others, the resulting RoP tests are correlated. Multiple-testing correction should therefore account for this dependency, for example, by using the effective-number-of-independent-tests approach [51,52] or a permutation test rather than applying a conservative Bonferroni adjustment. Additionally, methods developed for efficient epistasis screening [23,24,49,50] can be adapted to create a phase-aware RoP screening strategy that prioritizes variants for follow-up RoP testing.

LRS has enabled the observation of phase across long stretches of individual chromosomes. Our findings highlight the importance of phase-aware association analysis in fully interpreting the mechanisms of genetic variants uncovered by GWAS and for guiding follow-up functional studies; here, we provide a novel approach, the RoP, that can implement this. The application of the RoP to two CF modifier loci informs follow-up functional investigations that are required to advance the development of targeted and effective diagnostic and therapeutic strategies in CF.

## Supporting information

**S1 Text. Linear relationships between genotypes and phasetypes.**
(PDF)

**S2 Text. The coefficients of phase effects for different phasetypes.**
(PDF)

**S3 Text. The haplotype frequency odds-ratio (OR) approximates additive *cis* effects.**
(PDF)

**S4 Text. Relationships between RoP, saturated test and 4 df interaction test.**
(PDF)

**S1 Fig. The power of haplotype regression.** (A) Marginal effect contributed by allele A; (B) *cis* effects and (C) *trans* effects. In each scenario, haplotypes were simulated for n = 1,000 individuals, with $P_A = 0.2$ and $P_B$ ranging from 0.1 to 0.5, D' = 0. A continuous phenotype

was simulated by a linear model, with coefficients and residual standard deviation selected to maintain a heritability $h^2 = 0.01$. The power of haplotype regression was then assessed by 500 iterations.
(TIFF)

**S2 Fig. The type I error when marginal effect occurred at locus A.** Additive (A)(D), dominant (B)(E) and recessive (C)(F) inheritance models were examined; (A)-(C): the variants are not in LD (D' = 0) and (D)-(F): in LD (D' = 0.8). Dashed lines correspond to a type I error of 0.05.
(TIFF)

**S3 Fig. Power to detect phase effects.** (A)(D) Dominant *cis* effects, (B)(E) recessive *cis* effect and (C)(F) dominant *trans* effects. (A)-(C): the variants are not in LD (D' = 0) and (D)-(F): in LD (D' = 0.8).
(TIFF)

**S4 Fig. Impact of phasing uncertainty on RoP.** (A) Both alleles are rare with MAF = 0.05; (B) one rare allele (MAF = 0.05) and one common allele (MAF = 0.2); (C) both alleles are common with MAF = 0.2. For each scenario, haplotypes were simulated for n = 5,000 individuals. A continuous outcome was generated under an additive *cis* effect with heritability $h^2 = 0.003$. Switch errors were introduced only in individuals heterozygous at both variants at rates of 5%–20%, with 0% (no switch error) as the benchmark. The power of detecting *cis* and T1E for *trans* effects at each error rate level were estimated from 2,000 iterations.
(TIFF)

**S1 Table. Summary of type I error results.** "Yes" corresponds to the correct type I error rate.
(PDF)

**S2 Table. Impact of phasing uncertainty on RoP.** Switch errors were introduced only in individuals heterozygous at both variants at rates of 5%–20%, with 0% as the benchmark (no phasing error). Haplotypes were simulated for n = 5,000 individuals under an additive *cis* effect with heritability $h^2 = 0.003$. Power for detecting *cis* effects and type I error (T1E) for *trans* effects were estimated from 2,000 iterations per error level under three MAF scenarios: Rare-Rare (MAF = 0.05 for both alleles), Rare-Common (MAF = 0.05 and 0.2), and Common-Common (MAF = 0.2 for both alleles). RoP retained over 95% of its original *cis* power and maintained nominal *trans* T1E up to 10% switch error, with substantial T1E inflation at 20%.
(PDF)

## Author contributions

**Conceptualization:** Gengming He, Lisa J. Strug.

**Data curation:** Gengming He.

**Formal analysis:** Gengming He.

**Funding acquisition:** Lisa J. Strug.

**Investigation:** Gengming He.

**Methodology:** Gengming He.

**Resources:** Lisa J. Strug.

**Software:** Gengming He.

**Supervision:** Stephen W. Scherer, Lisa J. Strug.

**Visualization:** Gengming He.

**Writing – original draft:** Gengming He.

**Writing – review & editing:** Stephen W. Scherer, Lisa J. Strug.

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
