## [Decision Letter · Decision Letter 0]

27 Jul 2025

PGENETICS-D-25-00486

On the analysis of genetic association with long-read sequencing data

PLOS Genetics

Dear Dr. Strug,

Thank you for submitting your manuscript to PLOS Genetics. After careful consideration, we feel that it has merit but does not fully meet PLOS Genetics's publication criteria as it currently stands. Therefore, we invite you to submit a revised version of the manuscript that addresses the points raised during the review process.

Please submit your revised manuscript within 30 days Aug 26 2025 11:59PM. If you will need more time than this to complete your revisions, please reply to this message or contact the journal office at plosgenetics@plos.org. Please include the following items when submitting your revised manuscript:

We look forward to receiving your revised manuscript.

Kind regards,

Heather J Cordell

Academic Editor

PLOS Genetics

Xiaofeng Zhu

Section Editor

PLOS Genetics

Aimée Dudley

Editor-in-Chief

PLOS Genetics

Anne Goriely

Editor-in-Chief

PLOS Genetics

**Journal Requirements:**

3) We notice that your supplementary Figures, Tables, and information are included in the manuscript file. Please remove them and upload them with the file type 'Supporting Information'. Please ensure that each Supporting Information file has a legend listed in the manuscript after the references list.

**Reviewers' comments:**

Reviewer's Responses to Questions

Reviewer #1: A very readable paper.

The authors address an issue that is likely to be of increasing interest in particular in the context of understading the role of enhancer and superenhancers and the modulation of pleotropic effects.

The paper propose and explore a model. Perhaps a few words on the implementation would be desirable.

The comparison with existing approaches appear fair and the results plausibly explained

The authors focus on LRS. Pehaps some comments on adapting the framework to accomodate phase uncertainty iwould be of interest, It could facilitate the use of short read sequences.

Some typos could be corrected/clrified (e.g. page 31 after line 2 "++", or page 35 after line 1 "aa")

Reviewer #2: Summary

The authors propose a novel Regression on Phase (RoP) approach to detect phase effects of genetic variants on complex diseases, leveraging long-read sequencing (LRS) data. They demonstrate the method’s performance through simulation studies and application to cystic fibrosis (CF), revealing phase-dependent mechanisms that traditional GWAS approaches fail to capture. This is the first statistical method specifically developed to analyze both cis and trans phase effects in genetic association studies using LRS data. The methodology is well-presented, statistically sound, clearly explained, and supported by simulations and real data examples.

Comments

1. Theorem 1 (line 9): Should X_P be X_P^1 instead? Please clarify the notation.

2. The authors state that “The RoP test is designed to focus on testing the significance of the phase effects rather than the direction of the effects.” Can the authors elaborate on how the regression coefficients should be interpreted under this framework?

3. Figure 2: Can the authors provide more explanation as to why RoP is sensitive to cis and trans phase effects but not to allelic heterogeneity? Additionally, why is T1E=0.05 included on the plot? It would be helpful if the authors also comment on the performance of the non-conditional method for comparison.

4. Usability and scope: The method appears not suitable for large-scale or whole-genome analysis but rather for testing specific variants. Since exhaustive searches for epistasis signals across the genome are outside the intended scope, should multiple testing adjustments be considered when applying RoP? It would be useful if the authors could provide practical guidance on when and how to best apply the RoP test.

Furthermore, the absence of software implementation limits user accessibility. Can the authors clarify the recommended workflow? For example, should users first perform GWAS to identify variants of interest and then apply RoP? Are users expected to code genotype and phase data themselves? The development of a user-friendly tool would greatly facilitate adoption of this method.

Reviewer #3: The work introduced a novel regression on phase (RoP) method by encoding phase information into cis and trans. It can directly model and test the cis and trans effects of multiple genetic variants at a given genetic locus. Unlike traditional GWAS, which usually treat genetic variants as independent signals, RoP models can distinguish cis and trans phase effects between pairs of variants. Overall I find this manuscript well written with clearly presented results. I have a few specific comments:

1. The authors have provided examples of biallelic pairs, while in real world, multiple allelic variants may play a role together. The authors have stated that the RoP method can be applied in multi-allelic situation in the discussion section, but might lack a detailed description of how to extend into multi-allelic cases. It would be good if the authors can provide additional guidelines on how to further extend the method.

2. The authors applied the RoP methods to two loci and tested two non-phase and cis-phase cases. However, genome-wide analysis is missing, that is, whether the method can be computationally efficient and scalable. If the authors can wrap up the codes into a package, it would be beneficial to the community. At the same time, a description of suggested framework of how to apply RoP at genome-wide level would be helpful.

3. There have been haplotype regression methods available, which also include phase information to test the effects of each haplotype. Although the authors have included a haplotype OR test in simulations, to answer the question that whether RoP provides added value over simply modeling haplotypes categorically, it would be great to include a more explicit comparison to full haplotype regression and provide a comparison example on one real locus.

4. Phasing might be imperfect and misassigning haplotypes across variants can occur. If phasing errors occur, it can completely flip the assignment, turning cis into trans, and vice versa. It would be good to test the robustness in simulation studies by introducing different switch error rates.

5. The effect size terms for cis and trans are novel and biologically meaningful. However, beyond their significance, the magnitude of these effect sizes also needs to be discussed. The authors might use several simulated or real examples to showcase what the predicted phenotype means for each phase type count, and compare the effect sizes with models that include haplotype effects or genotype effects only.

**Have all data underlying the figures and results presented in the manuscript been provided?**

Reviewer #1: Yes

Reviewer #2: Yes

Reviewer #3: Yes

PLOS authors have the option to publish the peer review history of their article (what does this mean?). If published, this will include your full peer review and any attached files.

Reviewer #1: No

Reviewer #2: No

Reviewer #3: No

**Figure resubmission:**
---

## [Decision Letter · Decision Letter 1]

18 Sep 2025

Dear Dr Strug,

We are pleased to inform you that your manuscript entitled "On the analysis of genetic association with long-read sequencing data" has been editorially accepted for publication in PLOS Genetics. Congratulations!

Yours sincerely,

Heather J Cordell

Academic Editor

PLOS Genetics

Xiaofeng Zhu

Section Editor

PLOS Genetics

Aimée Dudley

Editor-in-Chief

PLOS Genetics

Anne Goriely

Editor-in-Chief

PLOS Genetics

Comments from the reviewers (if applicable):

Reviewer #2:

Reviewer #3:

Reviewer's Responses to Questions

**Comments to the Authors:**

Reviewer #2: Thank the authors for addressing my comments and questions in the revised manuscript. I have no further concerns as this time.

Reviewer #3: The authors have done a great job to address all my questions. I have no further comments.

**Have all data underlying the figures and results presented in the manuscript been provided?**

Reviewer #2: Yes

Reviewer #3: Yes

PLOS authors have the option to publish the peer review history of their article (what does this mean?). If published, this will include your full peer review and any attached files.

Reviewer #2: No

Reviewer #3: No

**Data Deposition**

http://datadryad.org/submit?journalID=pgenetics&manu=PGENETICS-D-25-00486R1

**Press Queries**

---

## [Editor Report · Acceptance letter]

PGENETICS-D-25-00486R1

On the analysis of genetic association with long-read sequencing data

Dear Dr Strug,

We are pleased to inform you that your manuscript entitled "On the analysis of genetic association with long-read sequencing data" has been formally accepted for publication in PLOS Genetics! Your manuscript is now with our production department and you will be notified of the publication date in due course.

With kind regards,

Anita Estes

PLOS Genetics

On behalf of:
